# Vitamin D Supplementation and Allergic Rhinitis: A Systematic Review and Meta-Analysis

**DOI:** 10.3390/medicina61020355

**Published:** 2025-02-18

**Authors:** Kei Kawada, Chiemi Sato, Tomoaki Ishida, Yui Nagao, Takaaki Yamamoto, Kohei Jobu, Yukihiro Hamada, Yuki Izawa Ishizawa, Keisuke Ishizawa, Shinji Abe

**Affiliations:** 1Department of Clinical Pharmacy Practice Pedagogy, Tokushima University Graduate School of Biomedical Sciences, Tokushima 770-8505, Japan; kawada.kei@tokushima-u.ac.jp (K.K.); sato.chiemi@tokushima-u.ac.jp (C.S.); c401904027@tokushima-u.ac.jp (Y.N.); yamamoto.takaaki@tokushima-u.ac.jp (T.Y.); ashinji@tokushima-u.ac.jp (S.A.); 2Department of Clinical Pharmacology and Therapeutics, Tokushima University Graduate School of Biomedical Sciences, Tokushima 770-8503, Japan; yuki.ishizawa@gmail.com (Y.I.I.); ishizawa@tokushima-u.ac.jp (K.I.); 3Department of Pharmacy, Kochi Medical School Hospital, Nankoku 783-8505, Kochi, Japan; jm-kouheij@kochi-u.ac.jp (K.J.); hamada_yukihiro@kochi-u.ac.jp (Y.H.); 4Department of General Medicine, Taoka Hospital, Tokushima 770-0941, Japan; 5Department of Pharmacy, Tokushima University Hospital, Tokushima 770-8503, Japan; 6Clinical Research Centre for Developmental Therapeutics, Tokushima University Hospital, Tokushima 770-8503, Japan

**Keywords:** allergic rhinitis, vitamin D, 25-hydroxyvitamin D, dietary supplements

## Abstract

*Background and Objectives*: Vitamin D supplementation is effective for allergic rhinitis; however, its usefulness is unclear. We conducted a systematic review and meta-analysis to examine the conditions in which vitamin D supplementation was effective in allergic rhinitis management. *Materials and Methods*: Randomized controlled trials of vitamin D supplementation used for patients with allergic rhinitis were searched for across different databases. We extracted scores on patients’ symptoms and the medication types used as the baseline treatments and performed a meta-analysis to evaluate the effect of vitamin D supplementation on allergic rhinitis symptoms. Meta-regression and subgroup analyses were performed for the average age, proportion of female participants, concomitant medications, vitamin D administration durations, and baseline serum 25-hydroxyvitamin D levels. *Results*: In total, 2389 articles were screened, and 5 randomized controlled trials (RCTs) were included in the meta-analysis. Compared with placebos, vitamin D supplementation alleviated allergic rhinitis symptoms, although the difference was not significant; there was significant heterogeneity among studies (standardized mean difference [SMD] = −2.69, 95% confidence interval [CI]: −6.20 to 0.82, I2 = 98%, *p* < 0.01). The proportion of female participants in the RCTs (slope: 0.21, *p* = 0.026) and concomitant corticosteroid use (slope: −9.16, *p* = 0.005) influenced the vitamin D response. Compared with the placebos, vitamin D supplementation without corticosteroids alleviated the allergic rhinitis symptoms (SMD = −0.56, 95% CI: −0.90 to −0.23). Combination treatment with corticosteroids also non-significantly alleviated symptoms. Additionally, the heterogeneity between studies was significant (SMD = −5.97, 95% CI: −13.55 to 1.16, I2 = 99%, *p* < 0.01). *Conclusions*: The study results suggest that vitamin D supplementation alleviates allergic rhinitis symptoms, although the effects differ according to the patient’s sex and concomitant medications.

## 1. Introduction

Allergic rhinitis (AR) is a common condition characterized by symptoms such as sneezing, itching, congestion, and a runny nose. It affects individuals of all ages and is often associated with other allergic diseases, such as asthma. The prevalence of AR has increased worldwide over the past few decades, affecting up to 40% of the population in developed countries and showing an increasing trend in developing countries [1,2]. AR can seriously affect quality of life, leading to poor sleep, poor academic performance, and reduced work productivity [3,4]. Treatment of AR includes drug therapy, immunotherapy, and allergen avoidance, and a combination of these treatment methods is used according to disease severity [5,6].

Vitamin D is known for its essential role in calcium metabolism and bone health [7,8]; however, accumulating evidence suggests its critical involvement in immune system regulation [9]. Vitamin D exerts immunomodulatory effects through its active form, 1,25-dihydroxyvitamin D, which interacts with the vitamin D receptor expressed in various immune cells [10]. In the innate immune system, vitamin D enhances the antimicrobial response by inducing the production of cathelicidins and defensins, which contribute to pathogen clearance [11]. Additionally, it modulates dendritic cell maturation, promoting a more tolerogenic phenotype that reduces excessive immune system activation [12]. In the adaptive immune system, vitamin D plays a crucial role in balancing pro-inflammatory and anti-inflammatory responses [13]. It promotes regulatory T cell (Treg) differentiation, which helps suppress inflammatory responses and maintain immune homeostasis [14]. Moreover, vitamin D inhibits the production of pro-inflammatory cytokines such as interleukin (IL)-6, tumor necrosis factor-α, and IL-17 while enhancing the secretion of anti-inflammatory cytokines like IL-10 [15,16,17]. This dual action is particularly relevant in allergic and inflammatory conditions, including AR, where excessive immune activation contributes to disease pathogenesis [18]. Furthermore, vitamin D supplementation enhances the efficacy of conventional AR therapies, such as antihistamines, steroids, and immunotherapy, supporting symptom control and improved quality of life [19].

Vitamin D supplements may be an effective adjunctive therapy for AR [16,19,20,21]. However, although clinical studies have suggested that concomitant vitamin D therapy may be effective, individual studies vary in terms of conditions and degree of efficacy, and the usefulness of vitamin D supplementation remains unclear. Thus, this study aimed to examine the specific conditions for the effectiveness of concomitant vitamin D supplementation on AR using a systematic review and meta-analysis methods.

## 2. Materials and Methods

### 2.1. Literature Search

This systematic review was conducted according to the Cochrane Handbook [22]. We followed the Preferred Reporting Items for Systematic Reviews and Meta-Analyses (PRISMA) statement [22]; the PubMed, Scopus, and Cochrane Library databases were searched for eligible studies from inception to 13 May 2024, without language restrictions. Details of the search strategy can be found in Appendix A. The search was supplemented by manual searches of reference lists of included publications and previous systematic reviews. This systematic review has not been pre-registered with public registries such as PROSPERO or OSF. However, the datasets and protocols generated and/or analyzed in this study are available from the corresponding author upon reasonable request, further information can be found in Appendix A.

### 2.2. Eligibility Criteria and Study Selection

Two researchers (T.I. and Y.N.) independently screened the titles and abstracts in the first stage and critically reviewed the full texts of the articles to assess eligibility in the second stage. Any discrepancies were resolved by discussion or consultation with a third, experienced, researcher (K.K.). Articles written in a language other than English were translated into English and assessed for eligibility according to our inclusion criteria. The inclusion criteria were as follows: (1) clinical studies meeting the following PICOs: (i) P (Population): patients with baseline treatment for AR; (ii) I (Intervention): vitamin D supplementation; (iii) C (Control): placebo or no supplementation; and (iv) O (Outcome): change in rhinitis symptoms; and (2) randomized control trials (RCTs) or equivalent controlled trials.

### 2.3. Data Extraction

Two researchers (T.I. and Y.N.) independently extracted data. Data from each study were tabulated and verified by a third researcher (K.K.) before inclusion in the analysis. Details of the data extraction are presented in Appendix B. The endpoint for AR was the symptom score. Mean ± standard deviation (SD) values were extracted from each study. For studies where SDs could not be obtained, the SDs were calculated based on the medians and interquartile ranges, as recommended by Wan et al. [23]. If data were missing or the reporting format was not suitable for a meta-analysis, the study authors were contacted via email or other methods of reporting as recommended in the Cochrane Handbook [22]. Calculations were performed from the data.

### 2.4. Quality of Assessment

The Jadad scale was used to assess the methodological and reporting quality of the RCTs [24].

### 2.5. Statistical Analyses

We used standardized mean differences (SMDs) with 95% confidence intervals (CIs) in the meta-analysis when different scales were used to report the same continuous outcome. Owing to the diversity of population characteristics, interventions, and outcome measures, a meta-analysis was performed using a random-effects DerSimonian–Laird model. Heterogeneity was assessed using the I^2^ statistic, with values > 50% suggesting considerable statistical heterogeneity [25]. Random-effects meta-regression was used to analyze the factors within the study that best explained the variance in SMDs. Meta-regression was used to analyze the contribution of the average age, proportion of female participants, concomitant medications, duration of vitamin D administration, and baseline serum 25-hydroxyvitamin D (25-(OH)D) level in terms of SMDs [26]. The significance threshold of all *p*-values was 0.05. All statistical analyses were performed using EZR version 1.51 (Saitama Medical Centre, Jichi Medical University, Saitama, Japan) [27].

## 3. Results

### 3.1. Systematic Review of the Literature

The PRISMA flow diagram showing the selection of studies for the meta-analysis of the effects of vitamin D on AR is shown in Figure 1. We identified 2389 records based on our search strategy and assessed the eligibility of 5 full-text articles after excluding 2384 articles. Five RCTs were included in the meta-analysis [1,16,28,29,30]. The characteristics of each included study are shown in Table 1. The five identified studies were all RCTs. A total of 370 patients were enrolled; the sample size ranged from 30 to 128. The age of the patients ranged from 5 to 70 years; four studies included patients older than 16 or 18 years, and only one study included patients aged 5–12 years. The treatment duration ranged from 4 weeks to 5 months. The dose of vitamin D ranged from 800 IU/day to 60,000 IU/week. The baseline treatment was subcutaneous allergen-specific immunotherapy, mometasone nasal spray, fluticasone nasal spray, antihistamine medication, and sublingual immunotherapy. Three RCTs evaluated the adverse effects of vitamin D supplementation; however, no serious adverse effects were reported. The countries where these studies were conducted and the sex distribution of patients are shown in Table 1.

### 3.2. Quality of the Studies

The results of the quality assessment of the included studies using the Jadad scale showed that two studies had a score of 5. The other studies had scores of 4, 2, and 1.

### 3.3. Overall Analysis

In the overall analysis of the five RCTs, vitamin D supplementation showed alleviation of AR symptoms compared with placebos; however, the difference was not significant (SMD = −2.69, 95% CI: −6.20 to 0.82, *p* = 0.134) (Figure 2). There was significant heterogeneity between studies (test for heterogeneity: I^2^ = 98%, *p* < 0.01) (Figure 2).

### 3.4. Meta-Regression

Meta-regression analysis showed that the proportion of female participants in the study and concomitant medications (concomitant use with corticosteroids) were significantly associated with SMDs (Table 2) (female participants: slope: 0.21, *p* = 0.026; concomitant use with corticosteroids: slope: −5.34, *p* = 0.015). However, the average age of the participants in the study (slope: −0.022, *p* = 0.62), duration of vitamin D administration (slope: 1.31, *p* = 0.15), and baseline serum 25-(OH)D levels (slope: −0.10, *p* = 0.44) were not associated with SMDs. Scatter plots in each meta-regression analysis are presented as Appendix A.

### 3.5. Subgroup Analysis with Concomitant Medications

Two RCTs involving the use of corticosteroids as concomitant medications [8,19] and three in which the concomitant medications were not corticosteroids were analyzed as subgroups [10,20,21]. Compared with the placebos, vitamin D supplementation without corticosteroids significantly alleviated AR symptoms (SMD = −0.56, 95% CI: −0.90 to −0.23, I^2^ = 0%, *p* = 0.97) (Figure 2). Combination with corticosteroids also alleviated AR symptoms, although the difference was not significant. Significant heterogeneity was found between studies (SMD = −5.97, 95% CI: −13.55 to 1.16, I^2^ = 99%, *p* < 0.01) (Figure 2).

## 4. Discussion

The meta-analysis of all eligible studies in this systematic review showed that, compared with placebos, vitamin D supplementation alleviated AR symptoms, although the heterogeneity was high. The meta-regression analysis indicated that concomitant corticosteroid use might have affected the heterogeneity. The subgroup analysis based on the results of the meta-regression analysis showed that vitamin D supplementation with medications other than corticosteroids significantly alleviated AR symptoms, whereas vitamin D supplementation with corticosteroids showed no significant improvement, probably because of the small number of studies included in the analysis and high level of heterogeneity.

Recent meta-analyses have reported the effect of vitamin D supplementation on AR in pregnant women, infants, and children [20,31,32]. Vitamin D supplementation in children significantly reduced AR symptom-medication scores, compared with a placebo [20]. Conversely, supplementation during pregnancy or infancy did not significantly reduce the risk of developing AR [31,32]. These reports suggest that vitamin D supplementation may have different effects in different target patients; furthermore, as no meta-analysis of vitamin D supplementation for adults with AR was reported, its effect was unknown.

Recently, the potential role of vitamin D supplementation in managing and preventing allergic diseases has attracted considerable interest. Various studies have investigated the effects of vitamin D on symptoms of asthma, atopic dermatitis, AR, and food allergies, especially in pediatric populations [19,21,31,32,33,34,35,36,37,38]. Vitamin D supplementation may be beneficial as an adjunctive therapy for steroid-resistant asthma [36] and significantly reduces the severity of atopic dermatitis in children [20,36]. Conversely, several other studies have reported insufficient evidence regarding the effectiveness of vitamin D supplementation in preventing or treating allergic diseases in infants, children, and adolescents, such as AR, asthma, food allergies, and atopic dermatitis [31,34,38]. Thus, evidence regarding the effectiveness of vitamin D supplements in preventing and treating allergic diseases is mixed and often inconclusive. In this study, we examined the specific conditions under which the effect of concomitant vitamin D therapy on patients with AR was observed by collecting and integrating the effects of vitamin D on AR in known studies. The results of this meta-analysis indicate that vitamin D supplementation improved AR symptoms significantly under certain conditions (concomitant use with medications other than corticosteroids). In addition, there did not appear to be any serious secondary effects of vitamin D supplementation, although these were not fully described in the reports included in this analysis (Table 1).

The findings of this meta-analysis suggest that vitamin D supplementation may be beneficial in alleviating AR symptoms, particularly when used in combination with treatments other than corticosteroids. Specifically, vitamin D has demonstrated potential synergistic effects when administered alongside subcutaneous allergen-specific immunotherapy, routine antihistamine medication, and sublingual immunotherapy. These results indicate that if vitamin D supplementation is to be integrated into AR management, it may be more advantageous to combine it with non-corticosteroid therapies. In addition to vitamin D, probiotics have been identified as potential adjunctive treatments for AR due to their immunomodulatory properties. Probiotics regulate gut microbiota composition and promote immune balance by increasing Treg levels and reducing pro-inflammatory cytokine production. Recent studies have suggested that probiotics, such as vitamin D, may enhance the efficacy of conventional AR treatments, further supporting their potential role in disease management [39]. Given the potential immunomodulatory synergy between these interventions, future studies should explore their combined effects to optimize therapeutic strategies.

Previous meta-analyses incorporating 12 trials to date have reported that allergen immunotherapy is equally or more effective than corticosteroids and other drug therapies in the treatment of AR [40]. Therefore, it is also expected that the effect of the baseline treatment was affected when vitamin D supplementation was combined with the baseline treatment in AR therapy. Corticosteroids are also a significantly effective pharmacotherapeutic option in the treatment of AR, especially for immediate symptomatic relief [41]. Therefore, the AR treatment effect of corticosteroids was so strong that the additive effect of vitamin D supplementation may have resulted in a large variability in the score. Furthermore, only two studies on the combination of vitamin D supplementation with corticosteroids were included in this analysis. However, because of the high heterogeneity of the two studies, they may not have been fully evaluated. In Guo et al.’s study, there was a marked improvement in AR symptoms with vitamin D supplementation [16]. In contrast, according to a study by Bhardwaj and Singh, the effect of vitamin D supplementation was mild [28]. In Guo et al.’s study, vitamin D was administered at a low dose of 800 IU/day (orally) for 4 weeks, whereas in the study by Bhardwaj and Singh, a higher dose of 60,000 IU/(oral) was administered for 4 weeks. Although higher doses of vitamin D are usually thought to produce stronger effects, Guo et al. showed stronger improvement, suggesting that other factors may have influenced the effects of vitamin D, not just the dose. Second, regarding differences in patient background, the participants in Guo et al.’s study were patients with moderate to severe AR (aged 16–60 years) and had relatively low serum 25-(OH)D levels at the beginning of the study, averaging 36.61 nmol/L. On the other hand, the study by Bhardwaj and Singh was limited to patients with AR and vitamin D deficiency (serum 25-(OH)D < 20 ng/mL); therefore, it is possible that the effect was not identified until sufficient amounts of vitamin D were supplemented. As described above, differences in study conditions, patient backgrounds, and the mechanism of action of vitamin D likely contributed to the heterogeneity among the studies. The choice of allergen immunotherapy or corticosteroids for AR treatment is expected to depend on patient-specific factors, such as severity of symptoms, convenience, and risk of adverse events. Therefore, vitamin D supplementation should be used in combination with adequate consideration of these factors.

The results of our meta-regression analysis suggest that the proportion of female participants in the included RCTs may significantly influence the effectiveness of vitamin D in AR. Specifically, trials with a lower proportion of female participants tended to show a greater treatment effect. This finding is consistent with previous research suggesting that sex hormones may modulate vitamin D metabolism and its immunoregulatory effects [42]. However, due to the limited number of studies included in our analysis, we were unable to determine an appropriate threshold for classifying studies into subgroups based on the proportion of female participants. As a result, we were unable to perform meaningful subgroup analysis. Further research with larger datasets is needed to validate these findings and to explore the potential mechanisms underlying sex differences in vitamin D efficacy.

This study has some limitations. First, only two RCTs of concomitant corticosteroid use were included in this analysis, and the heterogeneity of the two studies was high. This is most likely due to differences in patient background and treatment conditions between the two trials. Therefore, a random-effects model was used, and a subgroup analysis was attempted. However, only two studies were included in the corticosteroid combination group, which did not provide sufficient data for statistical examination. We also attempted to assess the influence of corticosteroid dosage on the effect of vitamin D. However, the two included RCTs that co-administered corticosteroids used different types and dosages of corticosteroids, making direct comparisons infeasible. Therefore, the results should be interpreted with caution. Further large-scale studies are needed to account for this effect. Second, articles on non-RCTs and RCTs with unclear evaluation criteria were excluded from the analysis because of the difficulty in assessing the risk of bias and extracting outcomes. Consequently, the number of cases was reduced, but the specificity of the analysis was improved, and the reliability of the results increased. Third, this study evaluated the improvement in total nasal symptom scores with vitamin D supplementation and was unable to evaluate individual symptom scores for rhinorrhea, nasal itching, sneezing, nasal congestion, red eyes, and itchy eyes. Only two of the included RCTs reported individual symptom scores. Other studies did not include these specific assessments and, therefore, could not be included in the pooled analysis. A more detailed assessment of individual symptoms would help clarify the exact effect of vitamin D on allergic rhinitis. Future studies should aim to incorporate these detailed symptom assessments to provide a more comprehensive evaluation of the therapeutic potential of vitamin D. Finally, studies on the effects of vitamin D supplementation in adults with AR remain limited and may not be available in a sufficient number of patients to validate its efficacy. Therefore, further clinical studies are required for accurate analysis.

In conclusion, vitamin D supplementation is expected to have clinical benefits in AR treatment, including alleviation of symptoms. However, at present, there is insufficient evidence to recommend concomitant vitamin D supplementation in AR treatment.

## Figures and Tables

**Figure 1 medicina-61-00355-f001:**
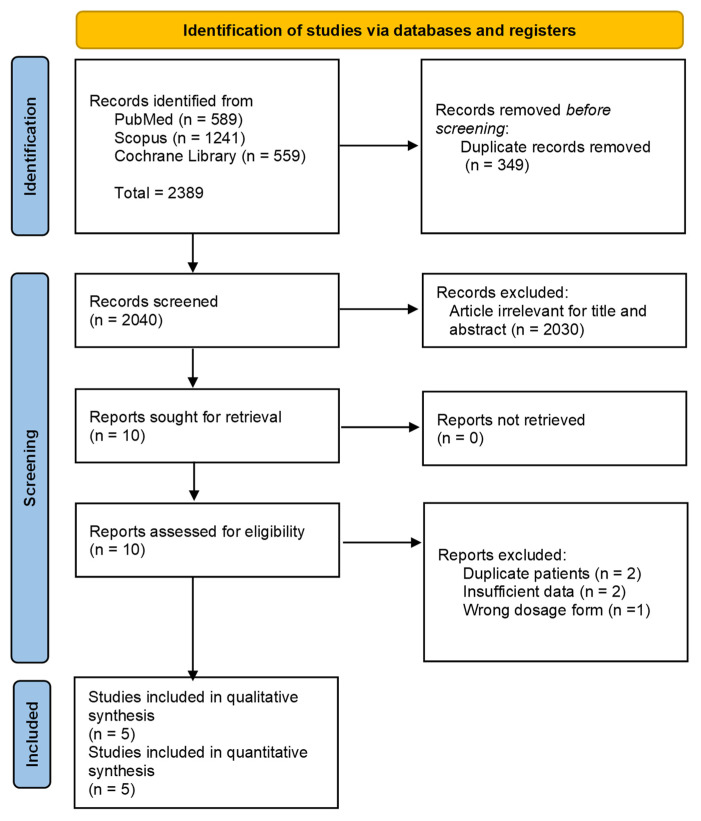
Flow diagram showing the process of study selection.

**Figure 2 medicina-61-00355-f002:**
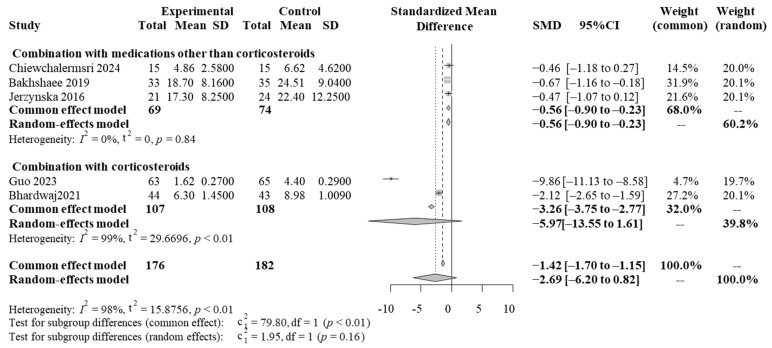
Forest plots showing the efficacy of vitamin D supplementation with concomitant medication for allergic rhinitis. SD, standard deviation; SMD, standardized mean difference. Chiewchalermsri 2024 [29], Bakhshaee 2019 [19], Jerzynska 2016 [30], Guo 2023 [16], Bhardwaj 2021 [28].

**Table 1 medicina-61-00355-t001:** Characteristics of the randomized controlled trials included in the meta-analysis.

Author/Year	Sample Size	Country	Study Design	Participants	Age (Years)	Sex (Female), n	Serum 25-(OH)D Levels at Baseline, Mean (SD)	Dose of Vitamin D	Baseline Treatment	Treatment Duration	Primary Outcome	Adverse Event
Chiewchalermsri et al. (2023) [29]	30	Thailand	RCT (randomized, double-blind, placebo-controlled clinical study)	Patients with HDM-allergic rhinitis	18–70	20	I: 20.3 (6.0) ng/mL C: 18.4 (5.6) ng/mL	Vitamin D2 (calciferol) 60,000 IU/week	SCIT mite mixed (1:1000) weekly	10 weeks	Symptom score	No serious adverse events related to vitamin D supplementation were reported.
Guo (2023) [16]	128	China	RCT (randomized, assessor/statistician-blinded, single-center trial with two parallel groups)	Patients with moderate-to-severe allergic rhinitis	16–60	39	I: 36.61 (3.58) nmol/L C: 36.40 (3.01) nmol/L	Oral vitamin D 800 IU/day	400 µg mometasone nasal spray	4 weeks	Symptom score	Intervention group: a mild dry nasal membrane (n = 10). No serious adverse reactions were observed in either of the groups.
Bhardwaj and Singh (2021) [28]	87	India	RCT (randomized controlled trial)	Patients with allergic rhinitis with vitamin D deficiency (serum 25-(OH)D levels < 20 ng/mL)	16–60	49	<20 ng/mL	Vitamin D3 cholecalciferol 60,000 IU/week	100 µg fluticasone nasal spray	4 weeks	Symptom score	NR
Bakhshaee et al. (2019) [19]	80	Iran	RCT (randomized, double-blind, placebo-controlled clinical trial)	Patients with allergic rhinitis with vitamin D deficiency (serum 25-(OH)D levels 10–20 ng/mL)	18–40	NR	I: 14 ng/mL C: 14.67 ng/mL	Weekly Peral 50,000 IU	Routine antihistamine medication (cetirizine)	8 weeks	Symptom score	NR
Jerzynska et al. (2016) [30]	45	Poland	RCT (prospective, randomized, double-blind, placebo-controlled study)	Patients with grass-related moderate-to-severe allergic rhinitis	5–12	19	I: 48.8 (24.3) ng/mL C: 43.3 (17.6) ng/mL	VD: 1000 IU/day 25 µg of cholecalciferol	SLIT five-grass pollens 300 (IR) daily	5 months	Symptom score	Sublingual itching (I: 45% versus C: 15.3%) was reported. There were no cases of termination of the study due to adverse events in any of the groups.

Abbreviations: C, control group; HDM, house dust mite; I, intervention group; IR, index of reactivity; IU, international unit; NR, not reported; RCT, randomized controlled trial; SCIT, subcutaneous alergen-specific immunotherapy; SD, standard deviation; SLIT, sublingual immunotherapy; VD, vitamin D.

**Table 2 medicina-61-00355-t002:** Summary of meta-regression analysis for concomitant medications, duration of vitamin D administration, and baseline 25-(OH)D levels.

	Slope	(95% CI)	*p*-Value
Age (years)	−0.022	−0.077 to 0.034	0.62
Female participants (proportion)	0.21	0.025 to 0.39	0.026
Concomitant steroid use	−5.34	−9.66 to −1.02	0.015
Duration of vitamin D administration (months)	1.31	−0.48 to 3.10	0.15
Baseline serum 25-(OH)D levels (ng/mL)	−0.10	−0.35 to 0.15	0.44

## Data Availability

This review was not pre-registered in a public registry such as PROSPERO or OSF. However, the datasets and protocol generated and/or analyzed in the current study are available from the corresponding author upon reasonable request.

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
