# Peer review of "Vitamin D Supplementation and Allergic Rhinitis: A Systematic Review and Meta-Analysis"

_medicina, 2025, doi:10.3390/medicina61020355_

Round 1

Reviewer 1 Report

Comments and Suggestions for Authors

In this study, Kawada et al. examined the effect of vitamin D on the symptoms of allergic rhinitis using publicly available data from randomized controlled trials. Their findings indicated that vitamin D improved the rhinitis symptoms in patients who did not use corticosteroids. However, they observed no significant difference in symptoms between the control group and the vitamin D-treated group among patients who were using corticosteroids. This is a relatively simple study, as it combines a small number of trials and only focuses on the total nasal symptom scale.

Specific comments:

1. The most significant concern regarding this study, as a meta-analysis, is the small number of trials included. In particular, the corticosteroid group combined two studies with high heterogeneity, which raises doubts about the adequacy of the conclusions drawn.

2. The authors concentrated solely on the total nasal symptom scale. It would be beneficial to consider other detailed symptom scales, such as those for sneezing, nasal itching, nasal congestion, and eye symptoms. 

3. Additionally, the authors could enhance their analysis by dividing the data according to factors like gender, age, and corticosteroid dosage. Without this deeper analysis (including the recommendation in comment #2), the study fails to provide new insights into the existing literature.

Author Response

Thank you for your careful review and helpful comments. We have thoroughly reviewed the entire manuscript and carefully responded to all comments. Our point-by-point responses are provided below:

Reviewer(s)' Comments to Author:

Reviewer: 1

Comments and Suggestions for Authors

In this study, Kawada et al. examined the effect of vitamin D on the symptoms of allergic rhinitis using publicly available data from randomized controlled trials. Their findings indicated that vitamin D improved the rhinitis symptoms in patients who did not use corticosteroids. However, they observed no significant difference in symptoms between the control group and the vitamin D-treated group among patients who were using corticosteroids. This is a relatively simple study, as it combines a small number of trials and only focuses on the total nasal symptom scale.

Specific comments:

  1. The most significant concern regarding this study, as a meta-analysis, is the small number of trials included. In particular, the corticosteroid group combined two studies with high heterogeneity, which raises doubts about the adequacy of the conclusions drawn.

Response to Reviewer:

Thank you for your valuable comments. We acknowledge that the small number of studies included in the meta-analysis, particularly in the corticosteroid subgroup, may limit the robustness of our conclusions. To address this, we conducted additional analyses to further investigate the impact of corticosteroid use on the effectiveness of vitamin D supplementation. However, given that only two studies were available in the corticosteroid subgroup, we were unable to perform a meaningful meta-regression analysis due to insufficient statistical power. We have now clarified this limitation in the Discussion section. Additionally, we have provided concrete examples of the differences in study conditions and patient backgrounds that may have contributed to the observed heterogeneity. Finally, we have revised the manuscript to emphasize the need for further large-scale randomized controlled trials to validate our findings. We hope that these revisions address your concerns.

Discussion (page 10, lines 240–257)

Before

However, because of the high heterogeneity of the 2 studies, they may not have been fully evaluated.

After

However, because of the high heterogeneity of the two studies, they may not have been fully evaluated. In Guo et al.’s study, there was a marked improvement in AR symptoms with vitamin D supplementation [16]. In contrast, according to a study by Bhardwaj and Singh, the effect of vitamin D supplementation was mild [28]. In the Guo et al.’s study, vitamin D was administered at a low dose of 800 IU/day (orally) for 4 weeks, whereas in the study by Bhardwaj and Singh, a higher dose of 60,000 IU/ (oral) was administered for 4 weeks. Although higher doses of vitamin D are usually thought to produce stronger effects, Guo et al. showed stronger improvement, suggesting that other factors may have influenced the effects of vitamin D, not just the dose. Second, as for differences in patient background, the participants in Guo et al.’s study were patients with moderate to severe AR (aged 16–60 years) and had relatively low serum 25-(OH)D levels at the beginning of the study, averaging 36.61 nmol/L. On the other hand, the study by Bhardwaj and Singh was limited to patients with AR and vitamin D deficiency (serum 25-(OH) D < 20 ng/mL; therefore, it is possible that the effect was not identified until sufficient amounts of vitamin D were supplemented. As described above, differences in study conditions, patient backgrounds, and the mechanism of action of vitamin D likely contributed to the heterogeneity among the studies.

Discussion (page 11, lines 273–283)

Before:

This study had some limitations. First, only 2 RCTs of concomitant corticosteroid use were included in this analysis, and the heterogeneity of the 2 studies was high.

After:

This study has some limitations. First, only two RCTs of concomitant corticosteroid use were included in this analysis, and the heterogeneity of the two studies was high. This is most likely due to differences in patient background and treatment conditions between the two trials. Therefore, a random-effects model was used, and a subgroup analysis was attempted. However, only two studies were included in the corticosteroid combination group, which did not provide sufficient data for statistical examination. We also attempted to assess the influence of corticosteroid dosage on the effect of vitamin D. However, the two included RCTs that co-administered corticosteroids used different types and dosages of corticosteroids, making direct comparisons infeasible. Therefore, the results should be interpreted with caution. Further large-scale studies are needed to account for this effect.

  1. The authors concentrated solely on the total nasal symptom scale. It would be beneficial to consider other detailed symptom scales, such as those for sneezing, nasal itching, nasal congestion, and eye symptoms.

Response to Reviewer:

Thank you for your valuable comment. Among the included RCTs, only two studies evaluated individual symptom components such as rhinorrhea, nasal itching, sneezing, nasal congestion, eye redness, and eye itching. However, the other RCTs did not report these individual outcomes, making it impossible to perform a pooled analysis. Evaluating these individual symptoms would provide a more detailed assessment of the effects of vitamin D. We agree that evaluating these individual symptoms would allow for a more detailed assessment of the effects of vitamin D. We have added information regarding this to the Discussion section as indicated below.

Discussion (page 11, lines 283–295)

Before:

Second, non-RCT articles and RCT articles with unclear evaluation criteria were excluded from the analysis because of the difficulty in assessing the risk of bias and extracting outcomes. Consequently, the number of cases was reduced, but the specificity of the analysis was improved, and the reliability of the results increased.

After:

Second, articles on non-RCTs and RCTs with unclear evaluation criteria were excluded from the analysis because of the difficulty in assessing the risk of bias and extracting outcomes. Consequently, the number of cases was reduced, but the specificity of the analysis was improved, and the reliability of the results increased. Third, this study evaluated the improvement in total nasal symptom scores with vitamin D supplementation and was unable to evaluate individual symptom scores for rhinorrhea, nasal itching, sneezing, nasal congestion, red eyes, and itchy eyes. Only two of the included RCTs reported individual symptom scores. Other studies did not include these specific assessments and, therefore, could not be included in the pooled analysis. A more detailed assessment of individual symptoms would help clarify the exact effect of vitamin D on allergic rhinitis. Future studies should aim to incorporate these detailed symptom assessments to provide a more comprehensive evaluation of the therapeutic potential of vitamin D.

  1. Additionally, the authors could enhance their analysis by dividing the data according to factors like gender, age, and corticosteroid dosage. Without this deeper analysis (including the recommendation in comment #2), the study fails to provide new insights into the existing literature.

Response to Reviewer:

Thank you for your insightful comment. In response to your suggestion, we examined the proportion of female patients and the average age in each RCT and conducted a meta-regression analysis to assess their impact on the effect size of vitamin D in allergic rhinitis. Our results indicate that the proportion of female participants in each study may significantly influence the effect of vitamin D on allergic rhinitis symptoms. In contrast, our meta-regression analysis did not show a significant relationship between the average age and the effect size of vitamin D, suggesting that age may not be a key determinant in its efficacy for allergic rhinitis treatment. Additionally, we attempted to evaluate the potential influence of corticosteroid dosage. However, the two studies that included corticosteroid co-administration used different corticosteroid types and dosing regimens, making direct comparisons of dosages impractical. As a result, we were unable to conduct a meaningful meta-regression analysis for corticosteroid dosage. To further substantiate our findings, we have updated the Results section to describe the outcomes of the meta-regression analysis in greater detail and have expanded Table 2 to include the results of the effects of sex and age on treatment responses. Additionally, we have added Figure S2 (Meta-Analytic Scatter Plot) to the Supplementary Materials to visually illustrate the relationship between these factors and effect size of vitamin D. In conjunction with the addition of these data, the necessary considerations have been added, as shown below.

Results (page 8, lines 160–168)

Before:

3.3. Meta-regression

Meta-regression analysis showed that concomitant medications (concomitant use with corticosteroids) were significantly and independently associated with SMDs (Table 2) (concomitant use with corticosteroids, slope: –9.16, P = 0.005). However, the duration of vitamin D administration was not associated with baseline serum 25(OH)D levels. Additionally, the duration of vitamin D administration (slope: –1.15, P = 0.152) and baseline serum 25(OH)D levels (slope: –0.21, P = 0.062) were not associated with SMDs.

After:

3.4. Meta-regression

Meta-regression analysis showed that the proportion of female participants in the study and concomitant medications (concomitant use with corticosteroids) were significantly associated with SMDs (Table 2) (female participants: slope: 0.21, P = 0.026; concomitant use with corticosteroids: slope: –5.34, P = 0.015). However, the average age of the participants in the study (slope: -0.022, P = 0.62), duration of vitamin D administration (slope: 1.31, P = 0.15), and baseline serum 25-(OH) D levels (slope: –0.10, P = 0.44) were not associated with SMDs. Scatter plots in each meta-regression analysis are presented as supplementary material (Figure S2).

Table 2

Before:

Slope

(95% CI)

P-value

Concomitant steroid use

Duration of vitamin D administration (months)

Baseline serum 25(OH)D level (ng/mL)

−9.16

−1.15

−0.21

−15.60 to −2.72

−2.73 to 0.43

−0.42 to 0.01

0.005

0.152

0.062

After:

Slope

(95% CI)

P-value

Age (years)

-0.022

 -0.077 to 0.034

0.62

Female participants (proportion)

0.21

 0.025 to 0.39

0.026

Concomitant steroid use

−5.34

 -9.66 to -1.02

0.015

Duration of vitamin D administration (months)

1.31

 -0.48 to 3.10

0.15

Baseline serum 25(OH) D level (ng/mL)

-0.10

 -0.35 to 0.15

0.44

Discussion (pages 10–11, lines 262–272)

Additionally stated:

The results of our meta-regression analysis suggest that the proportion of female participants in the included RCTs may significantly influence the effectiveness of vitamin D in AR. Specifically, trials with a lower proportion of female participants tended to show a greater treatment effect. This finding is consistent with previous research suggesting that sex hormones may modulate vitamin D metabolism and its immunoregulatory effects [42]. However, due to the limited number of studies included in our analysis, we were unable to determine an appropriate threshold for classifying studies into subgroups based on the proportion of female participants. As a result, we were unable to perform meaningful subgroup analysis. Further research with larger datasets is needed to validate these findings and to explore the potential mechanisms underlying sex differences in vitamin D efficacy.

Discussion (page 11, lines 278–281)

Additionally stated:

We also attempted to assess the influence of corticosteroid dosage on the effect of vitamin D. However, the two included RCTs that co-administered corticosteroids used different types and dosages of corticosteroids, making direct comparisons infeasible.

References

  1. Tomljenović, D.; Baudoin, T.; Megla, Ž.B.; Geber, G.; Scadding, G.; Kalogjera, L. Females have stronger neurogenic response than males after non-specific nasal challenge in patients with seasonal allergic rhinitis. Med Hypotheses 2018, 116, 114–118. https://doi.org/10.1016/j.mehy.2018.04.021.

Figure S2. Meta-Analytic Scatter Plot

Reviewer 2 Report

Comments and Suggestions for Authors

Very well organised paper.

I consider that topic is an important one nowadays, and I want to specify some specific mentions:

1. The introduction is a little bit too short. Maybe you can add some informations regarding the implications of Vitamin D in Immune System, more exactly something regarding the pathogenic mechanisms.

Author Response

Reviewer: 2

Comments and Suggestions for Authors

I consider that topic is an important one nowadays, and I want to specify some specific mentions:

  1. The introduction is a little bit too short. Maybe you can add some informations regarding the implications of Vitamin D in Immune System, more exactly something regarding the pathogenic mechanisms.

Response to Reviewer:

Thank you for your valuable comments. We appreciate your suggestion to expand the Introduction section by providing more details regarding the impact of vitamin D on the immune system, particularly regarding its pathogenic mechanisms. In response, we have revised the Introduction section to include a more comprehensive discussion of the immunomodulatory role of vitamin D. Specifically, we have added details on its regulation of innate and adaptive immune responses, including its influence on regulatory T cells (Tregs), dendritic cells, and pro-inflammatory cytokines (e.g., IL-6, TNF-α). These modifications enhance the background information, providing a clearer rationale for the study. We hope this revision meets your expectations, and we appreciate your insightful feedback.

Introduction (page 2, lines 52–69)

Before:

Vitamin D is a hormone with multiple actions, including promoting calcium absorption, regulation of bone metabolism, immune function support, and maintenance of muscle function [7]. Vitamin D supplementation in patients with AR has a significant effect on immunomodulation, including an increase in regulatory T-cell levels and a decrease in proinflammatory cytokine levels, such as interleukin-6 and tumor necrosis factor-β [8,9]. Furthermore, vitamin D supplementation enhances the efficacy of conventional AR therapies, such as antihistamines, steroids, and immunotherapy, supporting symptom control and improved quality of life [10].

After:

Vitamin D is known for its essential role in calcium metabolism and bone health [7,8]; however, accumulating evidence suggests its critical involvement in immune system regulation [9]. Vitamin D exerts immunomodulatory effects through its active form, 1,25-dihydroxyvitamin D, which interacts with the vitamin D receptor expressed in various immune cells [10]. In the innate immune system, vitamin D enhances the antimicrobial response by inducing the production of cathelicidins and defensins, which contribute to pathogen clearance [11]. Additionally, it modulates dendritic cell maturation, promoting a more tolerogenic phenotype that reduces excessive immune system activation [12]. In the adaptive immune system, vitamin D plays a crucial role in balancing pro-inflammatory and anti-inflammatory responses [13]. It promotes regulatory T cell (Treg) differentiation, which helps suppress inflammatory responses and maintain immune homeostasis [14]. Moreover, vitamin D inhibits the production of pro-inflammatory cytokines such as interleukin (IL)-6, tumor necrosis factor-α, and IL-17 while enhancing the secretion of anti-inflammatory cytokines like IL-10 [15–17]. This dual action is particularly relevant in allergic and inflammatory conditions, including AR, where excessive immune activation contributes to disease pathogenesis [18]. Furthermore, vitamin D supplementation enhances the efficacy of conventional AR therapies, such as antihistamines, steroids, and immunotherapy, supporting symptom control and improved quality of life [19].

References

  1. Cheng, L.; Chen, J.; Fu, Q.; He, S.; Li, H.; Liu, Z.; Tan, G.; Tao, Z.; Wang, D.; Wen, W.; Xu, R.; Xu, Y.; Yang, Q.; Zhang, C.; Zhang, G.; Zhang, R.; Zhang, Y.; Zhou, B.; Zhu, D.; Chen, L.; Cui, X.;, Deng, Y.; Guo, Z.; Huang, Z.; Huang, Z.; Li, H.; Li, J.; Li, W.; Li, Y.; Xi, L.; Lou, H.; Lu, M.; Ouyang, Y.; Shi, W.; Tao, X.; Tian, H.; Wang, C.; Wang, M.; Wang, N.; Wang, X.; Xie, H.; Yu, S.; Zhao, R.; Zheng, M.; Zhou, H.; Zhu, L.; Zhang, L. Chinese Society of Allergy Guidelines for Diagnosis and Treatment of Allergic Rhinitis. Allergy Asthma Immunol Res 2018, 10, 300–353. https://doi.org/10.4168/aair.2018.10.4.300
  2. Sly, R.M. Changing prevalence of allergic rhinitis and asthma. Ann Allergy Asthma Immunol 1999, 82, 233–252. https://doi.org/10.1016/s1081-1206(10)62603-8
  3. Schuler Iv, C.F.; Montejo, J.M. Allergic Rhinitis in Children and Adolescents. Pediatr Clin North Am 2019, 66, 981–993. https://doi.org/10.1016/j.pcl.2019.06.004
  4. Small, P.; Keith, P.K.; Kim, H. Allergic rhinitis. Allergy Asthma Clin Immunol 2018, 14, 51. https://doi.org/10.1186/s13223-018-0280-7
  5. May, J.R.; Dolen, W.K. Management of Allergic Rhinitis: A Review for the Community Pharmacist. Clin Ther 2017, 39, 2410–2419. https://doi.org/10.1016/j.clinthera.2017.10.006
  6. Portnoy, J.M.; Van Osdol, T.; Williams, P.B. Evidence-based strategies for treatment of allergic rhinitis. Curr Allergy Asthma Rep 2004, 4, 439–446. https://doi.org/10.1007/s11882-004-0009-1
  7. Holick, M.F. Vitamin D deficiency. N Engl J Med 2007, 357, 266–281. https://doi.org/10.1056/NEJMra070553
  8. Schrumpf, J.A.; van der Does, A.M.; Hiemstra, P.S. Impact of the Local Inflammatory Environment on Mucosal Vitamin D Metabolism and Signaling in Chronic Inflammatory Lung Diseases. Front Immunol 2020, 11, 1433. https://doi.org/10.3389/fimmu.2020.01433
  9. Aranow C. Vitamin D and the immune system. J Investig Med 2011, 59, 881–886. https://doi.org/10.2310/JIM.0b013e31821b8755

Reviewer 3 Report

Comments and Suggestions for Authors

Dear Authors,

I believe that your manuscript titled Vitamin D Supplementation and Allergic Rhinitis: A Systematic Review and Meta-Analysis addresses a topic of significant interest and is well-aligned with the objectives of the journal. Below, I have provided comments and suggestions:

INTRODUCTION
I suggest expanding this section to delve further into the topic of allergic rhinitis  and to provide a more detailed explanation of the underlying mechanisms of Vitamin D.

METHODS

  • This section is well-detailed and precise. I recommend including the PICO framework for your research and, if not already provided to the Editor, submitting the PRISMA checklist for reporting.
  • No registration protocol for PROSPERO or OSF is indicated. It would be beneficial to register this to ensure the methodological quality of your review.
  • The authors should also include as a supplementary file the detailed search strings used.

RESULTS

  • Figure 1: Please use the official PRISMA Flowchart figure.
  • Table 1: Include a legend for the acronyms used.
  • I believe it is important to mention that, prior to Table 1, a systematic review should describe the general characteristics of the included studies in text form, which is only partially addressed in section 3.2.
  • Additionally, there should be a section describing the quality of the studies and the risk of bias, assessed through the Jadad scale.

DISCUSSION
I believe the authors have thoroughly addressed this section. I suggest that lines 186-191 include a further discussion on other immunomodulatory substances. For example, you could explore how Vitamin D has a beneficial effect when combined with therapy for AR, similar to probiotics, which have also been shown to be an adjunctive treatment for allergic rhinitis (DOI: 10.52711/0974-360X.2023.00394 / DOI: 10.1097/MD.0000000000020095).

Please add a paragraph discussing the clinical implications of your study following the limitations section.

I believe that after these revisions, the manuscript will be suitable for potential publication.

Author Response

Reviewer: 3

Comments and Suggestions for Authors

I believe that your manuscript titled Vitamin D Supplementation and Allergic Rhinitis: A Systematic Review and Meta-Analysis addresses a topic of significant interest and is well-aligned with the objectives of the journal. Below, I have provided comments and suggestions:

INTRODUCTION

I suggest expanding this section to delve further into the topic of allergic rhinitis and to provide a more detailed explanation of the underlying mechanisms of Vitamin D.

Response to Reviewer:

Thank you for your valuable comments. We appreciate your suggestion to expand the Introduction section by providing more details regarding the impact of vitamin D on the immune system, particularly regarding its pathogenic mechanisms. In response, we have revised the Introduction section to include a more comprehensive discussion of the immunomodulatory role of vitamin D. Specifically, we have added details on its regulation of innate and adaptive immune responses, including its influence on regulatory T cells (Tregs), dendritic cells, and pro-inflammatory cytokines (e.g., IL-6, TNF-α). These modifications enhance the background information, providing a clearer rationale for the study. We hope this revision meets your expectations, and we appreciate your insightful feedback.

Introduction (page 2, lines 52–69)

Before:

Vitamin D is a hormone with multiple actions, including promoting calcium absorption, regulation of bone metabolism, immune function support, and maintenance of muscle function [7]. Vitamin D supplementation in patients with AR has a significant effect on immunomodulation, including an increase in regulatory T-cell levels and a decrease in proinflammatory cytokine levels, such as interleukin-6 and tumor necrosis factor-β [8,9]. Furthermore, vitamin D supplementation enhances the efficacy of conventional AR therapies, such as antihistamines, steroids, and immunotherapy, supporting symptom control and improved quality of life [10].

After:

Vitamin D is known for its essential role in calcium metabolism and bone health [7,8]; however, accumulating evidence suggests its critical involvement in immune system regulation [9]. Vitamin D exerts immunomodulatory effects through its active form, 1,25-dihydroxyvitamin D, which interacts with the vitamin D receptor expressed in various immune cells [10]. In the innate immune system, vitamin D enhances the antimicrobial response by inducing the production of cathelicidins and defensins, which contribute to pathogen clearance [11]. Additionally, it modulates dendritic cell maturation, promoting a more tolerogenic phenotype that reduces excessive immune system activation [12]. In the adaptive immune system, vitamin D plays a crucial role in balancing pro-inflammatory and anti-inflammatory responses [13]. It promotes regulatory T cell (Treg) differentiation, which helps suppress inflammatory responses and maintain immune homeostasis [14]. Moreover, vitamin D inhibits the production of pro-inflammatory cytokines such as interleukin (IL)-6, tumor necrosis factor-α, and IL-17 while enhancing the secretion of anti-inflammatory cytokines like IL-10 [15–17]. This dual action is particularly relevant in allergic and inflammatory conditions, including AR, where excessive immune activation contributes to disease pathogenesis [18]. Furthermore, vitamin D supplementation enhances the efficacy of conventional AR therapies, such as antihistamines, steroids, and immunotherapy, supporting symptom control and improved quality of life [19].

References

  1. Cheng, L.; Chen, J.; Fu, Q.; He, S.; Li, H.; Liu, Z.; Tan, G.; Tao, Z.; Wang, D.; Wen, W.; Xu, R.; Xu, Y.; Yang, Q.; Zhang, C.; Zhang, G.; Zhang, R.; Zhang, Y.; Zhou, B.; Zhu, D.; Chen, L.; Cui, X.;, Deng, Y.; Guo, Z.; Huang, Z.; Huang, Z.; Li, H.; Li, J.; Li, W.; Li, Y.; Xi, L.; Lou, H.; Lu, M.; Ouyang, Y.; Shi, W.; Tao, X.; Tian, H.; Wang, C.; Wang, M.; Wang, N.; Wang, X.; Xie, H.; Yu, S.; Zhao, R.; Zheng, M.; Zhou, H.; Zhu, L.; Zhang, L. Chinese Society of Allergy Guidelines for Diagnosis and Treatment of Allergic Rhinitis. Allergy Asthma Immunol Res 2018, 10, 300–353. https://doi.org/10.4168/aair.2018.10.4.300
  2. Sly, R.M. Changing prevalence of allergic rhinitis and asthma. Ann Allergy Asthma Immunol 1999, 82, 233–252. https://doi.org/10.1016/s1081-1206(10)62603-8
  3. Schuler Iv, C.F.; Montejo, J.M. Allergic Rhinitis in Children and Adolescents. Pediatr Clin North Am 2019, 66, 981–993. https://doi.org/10.1016/j.pcl.2019.06.004
  4. Small, P.; Keith, P.K.; Kim, H. Allergic rhinitis. Allergy Asthma Clin Immunol 2018, 14, 51. https://doi.org/10.1186/s13223-018-0280-7
  5. May, J.R.; Dolen, W.K. Management of Allergic Rhinitis: A Review for the Community Pharmacist. Clin Ther 2017, 39, 2410–2419. https://doi.org/10.1016/j.clinthera.2017.10.006
  6. Portnoy, J.M.; Van Osdol, T.; Williams, P.B. Evidence-based strategies for treatment of allergic rhinitis. Curr Allergy Asthma Rep 2004, 4, 439–446. https://doi.org/10.1007/s11882-004-0009-1
  7. Holick, M.F. Vitamin D deficiency. N Engl J Med 2007, 357, 266–281. https://doi.org/10.1056/NEJMra070553
  8. Schrumpf, J.A.; van der Does, A.M.; Hiemstra, P.S. Impact of the Local Inflammatory Environment on Mucosal Vitamin D Metabolism and Signaling in Chronic Inflammatory Lung Diseases. Front Immunol 2020, 11, 1433. https://doi.org/10.3389/fimmu.2020.01433
  9. Aranow C. Vitamin D and the immune system. J Investig Med 2011, 59, 881–886. https://doi.org/10.2310/JIM.0b013e31821b8755

METHODS

This section is well-detailed and precise. I recommend including the PICO framework for your research and, if not already provided to the Editor, submitting the PRISMA checklist for reporting.

No registration protocol for PROSPERO or OSF is indicated. It would be beneficial to register this to ensure the methodological quality of your review.

The authors should also include as a supplementary file the detailed search strings used.

Response to Reviewer:

Thank you for your thoughtful and constructive comments. We appreciate your valuable suggestions to enhance the clarity and methodological rigor of our systematic review and meta-analysis. Below, we address each of your recommendations in detail:

PICO Framework:

We appreciate your suggestion regarding the PICO framework. The PICO criteria for our systematic review are explicitly described in Section 2.2 (lines 91–96). Specifically, we have outlined the following:

P (Population): Patients receiving baseline treatment for allergic rhinitis.

I (Intervention): Vitamin D supplementation.

C (Control): Placebo or no supplementation.

O (Outcome): Change in allergic rhinitis symptoms.

Registration with PROSPERO or OSF:

We acknowledge the importance of registering systematic reviews to ensure transparency and methodological rigor. Unfortunately, our review was not pre-registered in a public registry such as PROSPERO or OSF. We recognize that registration enhances credibility and methodological transparency, and we will certainly consider this for future systematic reviews. We added the following to the Data Availability Statement.

Data Availability Statement: This review was not pre-registered in a public registry such as PROSPERO or OSF. However, the datasets and protocol generated and/or analyzed in the current study are available from the corresponding author upon reasonable request.

PRISMA Checklist Submission:

Thank you for highlighting the importance of adherence to PRISMA guidelines. To ensure compliance with the Systematic Review Reporting Standard, we have completed the PRISMA Checklist and will submit it as a supplemental document.

Search Strategy Documentation:

We have included a supplementary file containing the detailed search strings used in our literature search, as per your recommendation. These details can be found in Methods S1 of the Supplementary Materials.

Methods S1. Details of the search strategy

PubMed and Cochrane library

#1

"rhinitis, allergic, seasonal"[MeSH Terms] OR ("rhinitis"[All Fields] AND "allergic"[All Fields] AND "seasonal"[All Fields]) OR "seasonal allergic rhinitis"[All Fields] OR ("hay"[All Fields] AND "fever"[All Fields]) OR "hay fever"[All Fields] OR ("allergie"[All Fields] OR "hypersensitivity"[MeSH Terms] OR "hypersensitivity"[All Fields] OR "allergies"[All Fields] OR "allergy"[All Fields] OR "allergy and immunology"[MeSH Terms] OR ("allergy"[All Fields] AND "immunology"[All Fields]) OR "allergy and immunology"[All Fields]) OR ("allergic"[All Fields] OR "allergical"[All Fields] OR "allergically"[All Fields] OR "allergics"[All Fields] OR "allergization"[All Fields] OR "allergizing"[All Fields]) OR ("rhinitis"[MeSH Terms] OR "rhinitis"[All Fields] OR "rhinitides"[All Fields]) OR "rhinoconjunctivitis"[All Fields] OR ("rhinorrhea"[MeSH Terms] OR "rhinorrhea"[All Fields] OR "rhinorrhoea"[All Fields] OR "rhinorrheas"[All Fields]) OR ("anosmia"[MeSH Terms] OR "anosmia"[All Fields] OR "hyposmia"[All Fields]) OR ("anosmia"[MeSH Terms] OR "anosmia"[All Fields] OR "anosmias"[All Fields]) OR ("nasalance"[All Fields] OR "nasality"[All Fields] OR "nasalization"[All Fields] OR "nasalized"[All Fields] OR "nasally"[All Fields] OR "nose"[MeSH Terms] OR "nose"[All Fields] OR "nasal"[All Fields] OR "nasals"[All Fields])

#2

(((((("vitamin d"[MeSH Terms] OR "vitamin d"[All Fields] OR "ergocalciferols"[MeSH Terms] OR "ergocalciferols"[All Fields] OR ("cholecalciferol"[MeSH Terms] OR "cholecalciferol"[All Fields] OR "cholecalciferols"[All Fields] OR "colecalciferol"[All Fields]) OR ("hydroxycholecalciferols"[MeSH Terms] OR "hydroxycholecalciferols"[All Fields] OR "hydroxycholecalciferol"[All Fields]) OR ("25 hydroxyvitamin d"[Supplementary Concept] OR "25 hydroxyvitamin d"[All Fields] OR "25 hydroxyvitamin d"[All Fields] OR "calcifediol"[MeSH Terms] OR "calcifediol"[All Fields]) OR 25[UID]) AND ("hydroxide ion"[Supplementary Concept] OR "hydroxide ion"[All Fields] OR "oh"[All Fields])) AND ("vitamin d"[MeSH Terms] OR "vitamin d"[All Fields] OR "ergocalciferols"[MeSH Terms] OR "ergocalciferols"[All Fields])) OR 25[UID]) AND ("hydroxide ion"[Supplementary Concept] OR "hydroxide ion"[All Fields] OR "oh"[All Fields])) AND "D"[All Fields]) OR ("calcifediol"[MeSH Terms] OR "calcifediol"[All Fields] OR "calcidiol"[All Fields]) OR ("calcifediol"[MeSH Terms] OR "calcifediol"[All Fields]) OR ("dihydroxycholecalciferols"[MeSH Terms] OR "dihydroxycholecalciferols"[All Fields] OR "dihydroxycholecalciferol"[All Fields]) OR ("1 25 dihydroxyvitamin d"[Supplementary Concept] OR "1 25 dihydroxyvitamin d"[All Fields] OR "1 25 dihydroxyvitamin d"[All Fields]) OR ("calcitriol"[MeSH Terms] OR "calcitriol"[All Fields] OR "calcitriols"[All Fields]) OR ("ergocalciferols"[MeSH Terms] OR "ergocalciferols"[All Fields] OR "ergocalciferol"[All Fields]) OR ("vitamin d deficiency"[MeSH Terms] OR "vitamin d deficiency"[All Fields])

#3

"cohort"[All Fields] OR "cohort s"[All Fields] OR "cohorte"[All Fields] OR "cohorts"[All Fields] OR ("random allocation"[MeSH Terms] OR ("random"[All Fields] AND "allocation"[All Fields]) OR "random allocation"[All Fields] OR "randomization"[All Fields] OR "randomized"[All Fields] OR "random"[All Fields] OR "randomisation"[All Fields] OR "randomisations"[All Fields] OR "randomise"[All Fields] OR "randomised"[All Fields] OR "randomising"[All Fields] OR "randomizations"[All Fields] OR "randomize"[All Fields] OR "randomizes"[All Fields] OR "randomizing"[All Fields] OR "randomness"[All Fields] OR "randoms"[All Fields])

#1 and #2 and #3

Scopus

#1

(rhinitis and allergic and seasonal) OR (hay and fever) OR (allergie) OR hypersensitivity OR allergies OR allergy OR (allergy and immunology) OR allergic OR allergical OR allergically OR allergics OR allergization OR allergizing OR rhinitis OR rhinitides OR rhinoconjunctivitis OR rhinorrhea OR anosmia OR hyposmia OR anosmia OR anosmias OR nasalance OR nasality OR nasalization OR nasalized OR nasally OR nose OR nasal OR nasals

#2

(vitamin and d) OR ergocalciferols OR cholecalciferol OR cholecalciferols OR hydroxycholecalciferols OR hydroxycholecalciferol OR (25 and hydroxyvitamin and d) OR calcifediol OR calcifediol OR (25 and UID) OR dihydroxycholecalciferols OR (25 and dihydroxyvitamin and d) OR calcitriol OR calcitriols OR ergocalciferols OR ergocalciferol OR (vitamin and d and deficiency)

#3

cohort OR cohorts OR cohorte OR (random AND allocation) OR randomization OR randomized OR random OR randomisation OR randomisations OR randomize OR randomised OR randomising OR randomizations OR randomize OR randomizes OR randomizing OR randomness OR randoms

#1 and #2 and #3

RESULTS

Figure 1: Please use the official PRISMA Flowchart figure.

Response to Reviewer:

Thank you for your valuable feedback. In response to your comment, we have replaced Figure 1 with the latest official PRISMA flowchart diagram in accordance with the PRISMA 2020 guidelines.

Figure 1

Table 1: Include a legend for the acronyms used.

Response to Reviewer:

Thank you for your helpful comment. We have added an abbreviations list for Table1 as directed.

I believe it is important to mention that, prior to Table 1, a systematic review should describe the general characteristics of the included studies in text form, which is only partially addressed in section 3.2.

Additionally, there should be a section describing the quality of the studies and the risk of bias, assessed through the Jadad scale.

Response to Reviewer:

Thank you for your insightful comments and valuable suggestions. We appreciate your recommendation to enhance the description of the general characteristics of the included studies before Table 1 and to provide a dedicated section for assessing the quality of the studies and the risk of bias. In response, we have made the following revisions to improve the manuscript:

Description of General Characteristics of Included Studies:

We have added a detailed textual summary of the general characteristics of the included studies in Section 3.1 (lines 130–139).

Results (page 3, lines 130–139)

The five identified studies were all RCTs. A total of 370 patients were enrolled; the sample size ranged from 30 to 128. The age of the patients ranged from 5 to 70 years; 4 studies included patients older than 16 or 18 years, and only 1 study included patients aged 5–12 years. The treatment duration ranged from 4 weeks to 5 months. The dose of vitamin D ranged from 800 IU/day to 60,000 IU/week. The baseline treatment was subcutaneous allergen-specific immunotherapy, mometasone nasal spray, fluticasone nasal spray, antihistamine medication, and sublingual immunotherapy. Three RCTs evaluated the adverse effects of vitamin D supplementation; however, no serious adverse effects were reported. The countries where these studies were conducted and the sex distribution of patients are shown in Table 1.

Quality Assessment of Studies and Risk of Bias:

We have introduced a new Section 3.2: Quality of the Studies (lines 139–141), which outlines the assessment of study quality and risk of bias using the Jadad scale.

Results (page 4, lines 141–143)

3.2. Quality of the Studies

The results of the quality assessment of the included studies using the Jadad scale showed that 2 studies had a score of 5. The other studies had scores of 4, 2, and 1.

DISCUSSION

I believe the authors have thoroughly addressed this section. I suggest that lines 186-191 include a further discussion on other immunomodulatory substances. For example, you could explore how Vitamin D has a beneficial effect when combined with therapy for AR, similar to probiotics, which have also been shown to be an adjunctive treatment for allergic rhinitis (DOI: 10.52711/0974-360X.2023.00394 / DOI: 10.1097/MD.0000000000020095).

Please add a paragraph discussing the clinical implications of your study following the limitations section.

I believe that after these revisions, the manuscript will be suitable for potential publication.

Response to Reviewer:

Thank you for your valuable suggestion. In response to your comment, we have expanded the discussion in lines 186–191 to include other immunomodulatory substances. Specifically, we have added a discussion on probiotics, which have been reported to regulate gut microbiota and exhibit immunomodulatory effects similar to Vitamin D in the treatment of allergic rhinitis. The following text has been added to the manuscript:

Discussion (page 10, lines 223–230)

In addition to vitamin D, probiotics have been identified as potential adjunctive treatments for AR due to their immunomodulatory properties. Probiotics regulate gut microbiota composition and promote immune balance by increasing Treg levels and reducing pro-inflammatory cytokine production. Recent studies have suggested that probiotics, such as vitamin D, may enhance the efficacy of conventional AR treatments, further supporting their potential role in disease management [39]. Given the potential immunomodulatory synergy between these interventions, future studies should explore their combined effects to optimize therapeutic strategies.

References

  1. Ram, A.; Bhattacharjee, D.; Alam, S.M.; Jana, S.; Pal, R. A Review on the Resistance of Probiotic Microorganisms to Antibiotics. Asian J Pharm Technol 2024, 14, 330–0. https://doi.org/10.52711/2231-5713.2024.00054

Round 2

Reviewer 1 Report

Comments and Suggestions for Authors

The authors have addressed the concerns sincerely, which made the manuscript more informative.

Reviewer 3 Report

Comments and Suggestions for Authors

The authors have made substantial changes to the manuscript and it may be considered for publication.